# Diagnostic Accuracy of DNA-Methylation in Detection of Cervical Dysplasia: Findings from a Population-Based Screening Program

**DOI:** 10.3390/cancers16111986

**Published:** 2024-05-23

**Authors:** Narcisa Muresu, Mariangela V. Puci, Giovanni Sotgiu, Illari Sechi, Manuela Usai, Andrea Cossu, Marianna Martinelli, Clementina Elvezia Cocuzza, Andrea Piana

**Affiliations:** 1Department of Humanities and Social Science, University of Sassari, 07100 Sassari, Italy; manuelausai@hotmail.com; 2Clinical Epidemiology and Medical Statistics Unit, Department of Medicine, Surgery and Pharmacy, University of Sassari, 07100 Sassari, Italy; mvpuci@uniss.it (M.V.P.); gsotgiu@uniss.it (G.S.); 3Department of Medicine, Surgery and Pharmacy, University of Sassari, 07100 Sassari, Italy; illasechi@uniss.it (I.S.); andreacossu@uniss.it (A.C.); piana@uniss.it (A.P.); 4Department of Medicine and Surgery, University of Milano-Bicocca, 20900 Monza, Italy; marianna.martinelli@unimib.it (M.M.); clementina.cocuzza@unimib.it (C.E.C.)

**Keywords:** human papillomavirus, cervical cancer, DNA methylation, FAM19A4, miR124-2

## Abstract

**Simple Summary:**

The use of epigenetic biomarkers represents a promising tool in the diagnosis and prognosis of cancer. Several studies demonstrated the applicability and accuracy of DNA methylation analysis in the risk stratification of women with high-grade lesions, whereas lower evidence was reported in mild dyskaryosis and/or HPV-positive women. Particularly, the hypermethylation of host genes, FAM19A4 and miR124-2, is associated with an increase in the severity of cervical lesions and showed high accuracy in the detection of CIN2/3. These findings support the adoption of FAM19A4/miR124-2 as potential biomarkers for the prevention of cervical cancer and in reducing the over-referral to colposcopy examinations in a population-based screening program.

**Abstract:**

Background: Epigenetic biomarkers in cancer have emerged as promising tools for early detection, prognosis, and treatment response prediction. In cervical cells, hypermethylation of the host and viral HPV-genome increases with the severity of lesions, providing a useful biomarker in the triage of hr-HPV-positive women and during treatment. The present study focuses on evaluating the clinical performance of the FAM19A4/miR124-2 methylation test in a population-based cervical screening program. Methods: Previously collected cervical samples, after bisulfite-converted DNA, were analyzed by PreCursor-M+ kit (distributed by Fujirebio Europe), for DNA methylation. The sensitivity, specificity, and negative/positive predictive values of DNA methylation were compared to histology, colposcopy, the HPV-DNA test, and cytology results. Results: Among the 61-sample set, the specificity of methylation vs. positive histology (≥CIN2) and colposcopy (≥G2) were 87% and 90%, whereas the sensitivity was 50% and 33.3%, respectively. The combination of methylation analysis with standard methods increases diagnostic accuracy. Conclusions: Overall, we found a good specificity of DNA methylation in comparison to currently used techniques. Further larger studies could support the use of FAM19A4/miR124-2 as reliable biomarkers in the prevention of cervical cancer as triage in the screening protocol.

## 1. Introduction

Cancer poses a significant global challenge in the 21st century, representing a major burden on society, public health, and the economy. The current global cancer statistics from 2022 reveal nearly 20 million new cases and close to 10 million cancer-related deaths. Projections based on demographics suggest a significant increase, with an estimated 35 million new cancer cases annually expected by 2050 [1]. This increase underscores the urgent need for a comprehensive global approach to cancer control. Recent advancements in cancer prevention (e.g., innovative screening techniques and molecular biomarkers), diagnosis (e.g., liquid biopsies), and treatment (e.g., precision medicine and immunotherapies) have provided new insights into detecting and mitigating the impact of disease. For instance, cervical cancer screening programs have played a crucial role in decreasing the incidence rates of cervical cancer in many regions, particularly in Europe, Oceania, and Northern America. In 2022, more than 661,000 cases of cervical cancer and 348,000 deaths (4th most common cause of cancer-related death and incidence) were estimated globally [1]. The identification of the role played by high-risk human papillomavirus (hr-HPV) has allowed the implementation of primary (i.e., HPV-vaccination) and secondary preventive strategies (i.e., cervical screening). However, despite overall reductions, recent analysis shows an increasing risk among younger women in some countries, possibly due to changing sexual behaviors, inadequate screening uptake, and ineffective screening methods. Screening based on cytological examination decreased the epidemiological burden of cervical cancer, allowing early detection of precancerous cervical lesions and early therapy [2]. Compared to cytology, the detection of HPV-DNA has shown higher diagnostic rates for precancerous and cancerous cervical lesions, when compared with cytology, favoring its adoption as a first-line examination in screening protocols [3]. However, the replacement of cytological analysis by HPV-based screening has increased referrals to colposcopy, especially for women with borderline dyskaryosis as ASCUS (atypical squamous cell of undetermined significance) and LSIL (low-grade squamous intraepithelial neoplasia), attributed to the limited specificity of HPV-DNA test to discriminate between persistent and transient infections as well as to identify progress or regression of cervical lesions [4].

A reliable triage to reduce costs, unnecessary colposcopy, and overtreatment is needed, with several tools (e.g., HPV-16 and -18 genotyping, E6/E7 mRNA transcripts, p16/Ki-67 immunohistochemistry, and viral load) having been proposed until now [5,6,7]. In cervical cancer, hypermethylation of the host and viral genomes, which increases with the severity of lesions, could represent a biomarker to classify hr-HPV-positive women [8,9]. As reported for other tumors, methylation classifiers offer a promising approach for cancer diagnosis and classification, capturing the biological characteristics and clinical behavior of cancer [10]. DNA methylation mainly targets CpG dinucleotides, predominantly within CpG islands found in gene promoters. CpG islands, associated with 60% of human gene promoters, are typically unmethylated in normal cells, except for those linked to tissue differentiation. Cancer displays both global hypomethylation and localized hypermethylation, affecting tumor suppressor gene promoters and playing a crucial role in tumorigenesis [11].

Among the over one hundred human methylation biomarkers tested, approximately ten have emerged as clinically relevant for the diagnosis, prognosis, and treatment of cervical cancer, including CADM1, EPB41L3, FAM19A4, MAL, miR-124, PAX1, and SOX1 [12]. Large multicenter studies recommended the introduction of DNA methylation tests in cervical screening programs as a triage of HPV-positive women for the diagnosis of severe dysplasia and to predict the likelihood of disease progression or relapse in patients [9]. However, despite the early promising performance of methylation-based tests, their implementation in clinical practice still faces numerous limitations and is not yet effective. This may be attributed to the high heterogeneity in study settings, differences in targeted populations, sample types, and genes, and different methodologies in analytical protocols [2]. The current literature widely reports the accuracy of methylation tests in the detection of CIN2, CIN3, and cervical cancer [13]. However, scanty evidence is available regarding the use of methylation tests in population-based screening settings, where borderline or mild dyskaryosis results are the most prevalent diagnosis and the risk of overtreatment is higher.

The aim of the present study was to investigate the diagnostic and clinical performance of FAM19A4/miR124-2 methylation-based triage for severe cervical dysplasia in a population-based screening program. Particularly, we assess the diagnostic accuracy of methylation tests by comparing the results with standard methods involved in screening protocol (i.e., cytology, colposcopy, and histology). Furthermore, in order to best optimize the applicability of the test, the results obtained were compared with several possible triage scenarios within the screening pathway.

## 2. Materials and Methods

### 2.1. Study Design, Sample Selection, and Data Collection

An Italian cross-sectional, monocenter study was performed, considering samples previously collected from the regional cervical screening program.

Currently, the local cervical screening program consists of a preliminary cytology examination, followed by HPV-DNA test in cases of ASC-US (atypical squamous cells of undetermined significance) or LSIL (low-grade squamous intraepithelial lesion). In cases of positivity for at least one high-risk HPV-genotype (HR-HPV), women undergo colposcopy and histological examination for ≥1 grade.

From the LBC (liquid-based cytology) samples previously analyzed within the screening program, the study samples were randomly selected regardless of whether the HPV-genotype was detected based on cytological and histological results.

All demographic (i.e., age) and clinical (i.e., cytology, HPV-DNA test, colposcopy, and histology results) data were anonymously collected for each patient.

### 2.2. Sample Processing

The cytological examination was based on TBS-2001 criteria [11]. The HPV genotyping was carried out after nucleic acid extraction by QIAamp DNA Mini Kit (Qiagen, Hilden, Germany) [14], using the Anyplex II HPV HR detection kit (Seegene Inc., Seoul, Republic of Korea), allowing the detection of 14 HR-HPV genotypes, including HPV-16, 18, 31, 33, 35, 39, 45, 51, 52, 56, 58, 59, 66, and 68, as previously described [15,16].

### 2.3. DNA Isolation and Methylation Analysis

The DNA methylation assay was performed by the available commercial kit PreCursor-M+ (Fujirebio Europe N.V., Gent, Belgium) [17], a multiplex real-time PCR test that starts with bisulphite-converted DNA and identifies the level of methylation in promoters of host cell genes FAM19A4 and hsa-miR124-2 as biomarkers associated with the progression of cervical lesions, and simultaneously provides the amplification of the β-actine gene as an internal control.

For the PreCursorM+ Assay (distributed by Fujirebio Europe), 2 mL of PreservCyt cellular suspension were centrifuged for 5 min at a speed of 10,000× *g* and resuspended in 200 μL of PBS1x. After DNA extraction, the concentration of genomic DNA was measured with Qubit™ 4 Fluorometer Instrument (Thermo Fisher, Waltham, MA, USA) using Qubit 1X dsDNA BR Assay Kit (Thermo Fisher), an assay designed to measure genomic DNA concentration in a range of 4–4000 ng. Through the bisulphite reaction, up to 200 ng/45 μL of isolated genomic DNA was converted using the EZ DNA Methylation Kit (Zymo Research Europe, Irvine, CA, USA) [18].

To perform the PreCursorM+ Test, 17.5 μL of ready-to-use real-time PCR Master Mix and 2.5 μL of DNA, positive and NTC controls, were added in PCR instrument tubes. The multiplex PCR was run on a MIC IVD instrument (Bio Molecular Solutions, Port of Spain, Trinidad and Tobago). The samples were scored as valid when the Ct value of ACTB was ≤26.4, and hypermethylation status was considered positive if at least one of the methylation marker genes had a ΔΔCt below the cut-off. A sample was positive for methylation if the ΔΔCt value was ≤9.66 for FAM19A4 and/or ≤6 for has-miR124-2. The ΔΔCt values were calculated using a fixed calculation template provided by the distribution company (Fujirebio Europe, Gent, Belgium) [17].

### 2.4. Statistical Analysis

Sample characteristics were summarized by mean and standard deviation (SD), median and interquartile range (IQR), and absolute and relative (percentages) frequencies.

To evaluate the performance of the DNA methylation test, sensitivity, specificity, positive predictive values (PPV), and negative predictive values (NPV) were calculated according to conventional formulas for all clinical samples. DNA methylation results were compared with different techniques adopted in the screening program: cytology, the HPV-DNA test, colposcopy, and histology. Following the international classification system, we assumed as positive results a colposcopy grading of ≥1 and CIN2 and CIN3 histological classification [19]. Receiver operating characteristic (ROC) curve analysis was performed to evaluate the performance of the methylation test alone and combined with the HPV-DNA test in the identification of positive colposcopy cases (i.e., ≥1 G1). The results were reported as area under the ROC curve (AUC) measures and a 95% confidence interval (CI).

Differences in the performance of the methylation test were evaluated using Fisher’s exact test. A *p*-value < 0.05 was considered statistically significant. Statistical computation was carried out through STATA17 software.

## 3. Results

A total of 61 specimens were collected. According to the primary aim of the study, we calculated the specificity, sensitivity, NPV, and PPV of the methylation test compared with standard methods (i.e., cytology, HPV test, colposcopy, and histology) for each sample (N = 61). The demographic and clinical characteristics of the study population are reported in Table 1.

### 3.1. Clinical Sample Characterization

Fifty-five samples (90.2%) were positive for at least one HR-HPV genotype, and, among those, 60% showed multiple infections. The most prevalent genotype was HPV 16 (18%), followed by HPV-68, -66, with 11.5%, and -59, and -58 with 9.2% (Figure 1).

On cytological examination, most specimens were classified as LSIL (22/61; 36.1%), followed by ASCUS (21/61; 34.4%), whereas 29.5% of specimens were negative (18/61; 29.5%). As per screening protocol, 50 samples (82%) required further investigation by colposcopy, with the following results: 16 samples (32%) were negative, 4 (8%) classified as G0, 20 (40%) as G1, and 10 (20%) as G2. Moreover, for 46% (28/61) of the specimens analyzed, the result of the histological examination was available with the following results: most of the biopsies were negative (16; 57.1%), 8 (28.6%) CIN1, 3 (10.7%) CIN2, and 1 (3.6%) was classified as CIN3 (Appendix A).

### 3.2. Results of DNA Methylation Analysis

Diagnostic and clinical performance of DNA methylation were analyzed by comparing every single step of the screening process, which includes cytology analysis, HPV-DNA test, colposcopy, and histology, and assessing for all tests the sensitivity, specificity, and positive and negative predictive values (i.e., PPV and NPV).

DNA methylation analysis was positive in 19.7% (12/61) of samples, of which one showed positivity for both target genes, whereas the remaining 11 samples tested positive exclusively for the FAM19A4 gene. The ΔΔCq median (IQR) value observed for FAM19A4 was 9.3 (7.7–10.0), and 9.4 (8.7–10.5) for miR124-2. The results of the DNA methylation test in comparison with cytology, colposcopy, and histology analysis are reported in Figure 2.

The statistical analysis did not evidence any statistically significant differences between the results of the methylation test and standard techniques (Figure 2). However, it is underlined that the majority of specimens resulted negative on cytology, HPV-DNA, colposcopy (i.e., negative and G0), and histology (i.e., negative and CIN1) and were also negative on methylation testing, for 83%, 100%, 80%, and 87.5%, respectively.

Comparing the results of methylation analysis with colposcopy and histology examinations, we found a sensitivity of 33.3% and 50% and a specificity of 90% and 86.7%, respectively (Table 2).

The positive predictive value of DNA methylation testing, when compared with other diagnostic tests, was higher than 0.80, except for histology, where the PPV was 0.40. Conversely, the negative predictive value was higher if compared with histology data (0.90), but lower versus cytology, HPV-DNA, and colposcopy, with values of 0.33, 0.12, and 0.47, respectively (Table 2).

In the context of our study, we computed the sensitivity, specificity, positive predictive value (PPV), and negative predictive value (NPV) of cytology within the sample set to establish a benchmark for comparison with the currently employed protocol. The sensitivity of cytology ranged from 69.1 to 100.0 compared with HPV-DNA, colposcopy, and histology, whereas specificity ranged from ~17% to 55% (Table 3). Overall, we found a lower sensitivity and higher specificity when DNA methylation was compared with cytology.

We also considered the combination of the methylation test and HPV-DNA in the detection of high-grade lesions at colposcopy examination (i.e., ≥1 grade). We found a sensitivity of 31%, a specificity of 80.0%, and a PPV and NPV of 0.71 and 0.42, respectively (Table 4). Particularly, focusing on samples positive for HPV-16/-18, considered to be the main causes of cervical cancer occurrence, we report a higher sensitivity and specificity, with 42.9% and 83.3%, respectively.

The analysis of ROC curves showed an AUC value for methylation alone and an AUC for methylation and HPV-DNA cotesting, versus colposcopy, of 0.53 (95% CI: 0.41–0.65) and 0.54 (95% CI: 0.38–0.70), respectively (Appendix A).

Finally, we compared possible triage scenarios for HPV-positive women within the screening program (Table 5). Analysis of the sample set shows that single assays exhibited limitations in terms of accuracy and predictive power. In particular, the methylation test, or HPV-16 positivity, shows good specificity (80% and 75%) but low sensitivity (31% and 37%, respectively). On the other hand, the performance of cytology in diagnosing of ≥1 grade lesions at colposcopy shows acceptable results in terms of sensitivity (87%), but limited specificity (55%). In contrast, the combination of cytology with the methylation test significantly improved the specificity of detection of ≥1 grade at the colposcopy examination.

## 4. Discussion

The World Health Organization recently launched a global initiative to reduce the incidence of cervical cancer with the 90-70-90 strategies: 90% of girls < 15 years of age who are fully vaccinated, 70% of screened women in the targeted population, and 90% of precancer or cancerous lesions treated [20]. Moreover, WHO strongly recommends the use of HPV-DNA as a primary test in screening programs, considering its higher accuracy compared to cytology or visual inspection with acetic acid (VIA), besides the opportunity to be carried out on self-collection specimens [21,22,23]. Despite its higher sensitivity, HPV-DNA showed a low ability to distinguish transient from persistent infections, and, for this reason, numerous studies highlighted the need for new biomarkers for risk stratification in HPV-positive women to reduce over-referral to cytology and over-treatment.

Recent literature shows that aberrant epigenetic changes contribute to tumor initiation, progression, and metastasis by silencing tumor suppressor genes or activating oncogenes. In this pathway, epigenetic biomarkers offer valuable insights into the molecular mechanisms underlying cancer development and progression [24]. In fact, the occurrence of CpG island methylation can cause faulty gene expression and genomic instability, which is correlated with chromosome condensation and silencing of gene expression.

In cervical cells, hypermethylation of host and viral genomes seems to be a promising and noninvasive biomarker associated with cancer progression and disease severity. Particularly, FAM19A4 and hsa-miR124-2 have shown high sensitivity to detect cervical cancer and advanced CIN3 cervical lesions, with a positivity rate of 95% and 77%, respectively [9,24].

FAM19A4, belonging to the TAFA gene family, was initially detected in HPV-16 keratinocytes, where it was found to be downregulated in cervical cancers. Subsequently, growing evidence demonstrated a progressive increase in FAM19A4 methylation levels in squamous cell carcinomas compared to CIN3 lesions, as well as in adenocarcinomas compared to adenocarcinomas in situ [25]. Along with the function of DNA methylation, there is evidence for a role in the post-transcriptional regulation of miRNAs, noncoding RNA consisting of approximately 23 nucleotides. In particular, miRNA loci are subject to epigenetic regulation and exhibit a notable association with the HPV insertion site in cervical cancer, particularly in the more advanced stages of the disease [26,27].

In this scenario, our study was focused on the methylation status of promoter host cell genes FAM19A4 and hsa-miR124-2 in samples collected during the local cervical screening program and aimed to evaluate, for the first time in our setting, the potential adoption of methylation assays in the management of HPV-positive women and in the detection of severe cervical dysplasia.

Overall, a good specificity (>80%) of DNA methylation was found when compared to colposcopy (i.e., ≥G1) and histology (i.e., ≥CIN2), correctly discriminating cases without diseases, as confirmed by NPVs (0.91 for CIN2+ detection).

Several studies were focused on markers for the screening of CIN and invasive cancers. The FAM19A4/miR-124-2 methylation test demonstrated high sensitivity and specificity in gynecologic outpatients, regardless of sample types and geographical regions. Additionally, the FAM19A4/miR-124-2 methylation test showed an overall sensitivity of 77.2% and a specificity of 78.3% for the CIN3 case [25,28]. A recent systematic review that assessed the performance of DNA methylation for high-grade cervical lesions (i.e., CIN2+) showed a pooled specificity of 74% (ranged between 69–78% among study centers) and referred to the most frequently studied genes in cervical specimens (i.e., CADM1, FAM19A4, MAL, and miR124-2) [8]. Kong and colleagues [24] found a higher diagnostic accuracy of the methylation assay compared with hr-HPV testing in the detection of severe cervical lesions. [24]. Moreover, a large multicenter European study revealed a specificity of 78.3% for ≤CIN1 diagnosis among HPV-positive women [9]. Our study findings underscored the potential role of the methylation assay to detect true negative cases when compared with histology and colposcopy, as well as the higher specificity of the DNA methylation test versus cytology for CIN2+ (83.3% vs. 25%, respectively) and ≥G1 colposcopy (91% vs. 55%, respectively). Translated into clinical practice, these findings can reduce the overtreatment rate and unnecessary follow-up visits, which could raise healthcare costs and increase anxiety for patients. Moreover, methylation-based triage shows several advantages if compared with cytology-based approaches, such as objectivity (the operator-independent method), which is performed directly on DNA extracted for HPV-DNA testing and can be applied both to clinician- and self-collected specimens [12]. In fact, the FAM19A4/mir124-2 assay demonstrates similar clinical sensitivity and specificity for triage of HPV-positive women following self-sampling compared to other methodologies, regardless of the self-sampling device used (e.g., vaginal lavage and brush) [29,30].

Conversely, in our sample set, we found a lower detection rate for true positive samples when compared with colposcopy and histology (sensitivity of 33.3% and 50%, respectively), mainly related to the fact that the majority of samples were classified as mild- or low-grade cervical lesions. Our results are consistent with the scientific literature, which highlighted a high sensitivity of the methylation test for CIN2, or more severe disease, but lower for low and mild cervical lesions. A decreasing trend was described by the degree of dysplasia with a sensitivity of 48%, 43%, and 27% for CIN3 in the ASCUS, LSIL, and NILM groups, respectively [9]. Otherwise, we reported a higher sensitivity of standard techniques (i.e., cytology) in detections of patients with severe disease in comparison with histology and colposcopy, with a sensitivity of 100% and ~87% for CIN2+ and ≥G1 cases, respectively. The potential applicability of the methylation test in combination with other highly sensitive methods (e.g., the HPV-DNA test) should be considered to improve the sensitivity of the test in detecting grade ≥ 1 colposcopy lesions. In addition, further risk stratification could be achieved considering HPV-genotype, which has been suggested as a triage for HPV-positive women [27,31]. Our results confirmed the higher accuracy of the methylation assay when combined with cytology and HPV testing.

In addition to exploring the possible introduction of the methylation test in the triage of HPV-positive women, several studies investigated the prognostic role of methylation as a marker of treatment outcome. In a longitudinal study, negative results for FAM19A4/miR124-2 at baseline were associated with a regression of disease in untreated CIN2/3 [32]. Similar findings were shown in a screening cohort with 14 years of follow-up, where the risk of cervical cancer in those with baseline negative methylation is equal to that in those with negative cytology results [33]. Moreover, in shorter follow-up periods (i.e., 24 months), negative methylation was associated with clinical regression in women with CIN, providing a reliable tool for determining whether immediate treatment or a wait-and-see approach is warranted and as a monitor of treatment response [28,32].

Overall, a high specificity of the hypermethylation status of FAM19A4/miR124-2 promoter genes was found in our monocenter study, with a better performance than that of cytology. On the other hand, the low positivity rate of the methylation test in our sample set is related to the low prevalence of high-grade lesions observed at colposcopy and histological examination, consistent with previous reports [9,34]. Moreover, as recently demonstrated, further target genes could be considered to improve the performance of the methylation test [35].

The limitations of the present study are mainly related to the retrospective design, which can be associated with missing clinical information. However, we decided to exclude a histological endpoint, investigating the performance of the test in a real screening population where the number of women referred to histologic examination is usually low. Additionally, the small sample size, as well as the low positivity rate for high-grade lesions (e.g., CIN2+), could have influenced the sensitivity and specificity values of the methylation test and need to be evaluated to improve the number of samples.

Lastly, it is imperative to emphasize the necessity for a larger sample size to facilitate a comprehensive assessment of the diagnostic and prognostic role of methylation assay with a more nuanced understanding of how methylation patterns correlate with disease severity, demographic factors, and virological variables (e.g., age and HPV genotype), both in screening and colposcopy settings. Moreover, it is essential to underscore the importance of evaluating the prognostic value of methylation tests in predicting the outcomes of low-grade cervical lesions. By conducting longitudinal studies with extended follow-up periods, researchers can better discern the predictive capabilities of methylation assays in identifying low-grade lesions that are more likely to progress to high-grade or invasive disease. This information is crucial before considering the implementation of methylation testing in routine screening programs as an evidence-based intervention.

## 5. Conclusions

In conclusion, our findings support the potential role of DNA methylation in the triage of women within cervical cancer screening. The clinical and diagnostic accuracy of the test needs to be confirmed by enrolling in larger prospective studies and considering the key role of different epidemiological and clinical variables. Further research is needed to extend the evaluation of methylation as a triage strategy and inform the development of international validation guidelines, including technical specifications, reliable target genes, and the handling of invalid samples. Moreover, it is crucial to assess the diagnostic accuracy of the methylation assay combined with HPV-DNA and/or cytology analysis to optimize the allocation of public health resources for the prevention and treatment of cervical cancer on a global scale.

## Figures and Tables

**Figure 1 cancers-16-01986-f001:**
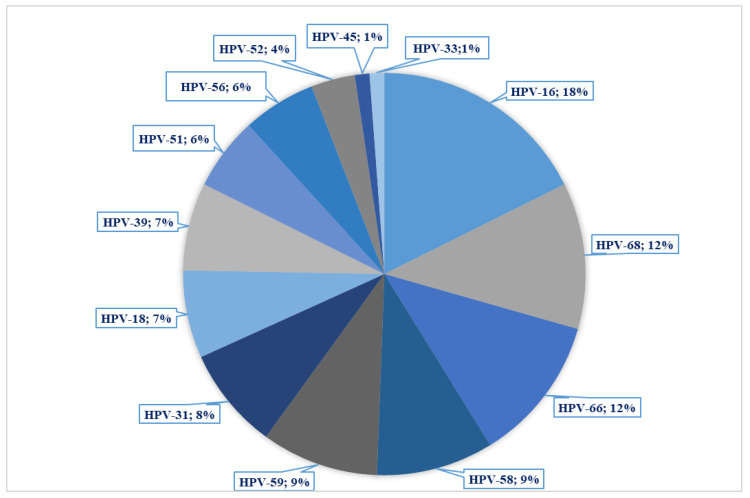
Distribution of HPV genotypes in the sample set.

**Figure 2 cancers-16-01986-f002:**
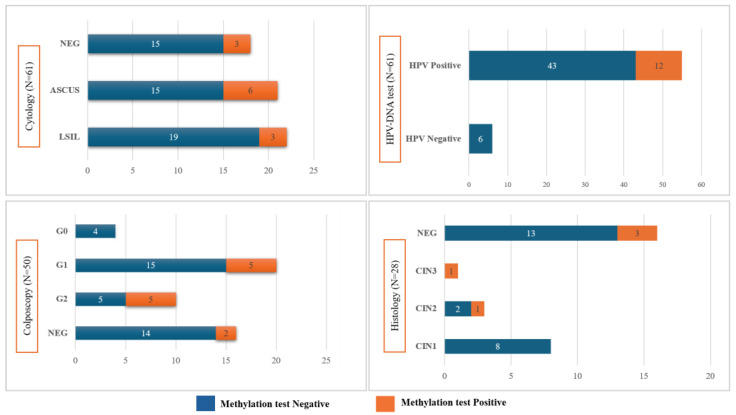
Results of the DNA methylation test of the sample study, stratified by cytology, HPV-DNA test, colposcopy, and histology analysis.

**Table 1 cancers-16-01986-t001:** Demographic and clinical characteristics of the study population (N = 61).

HPV positivity, *n* (%)	55 (90.2)
Mono infection, *n* (%)	22 (40.0)
Multiple infections, *n* (%)	33 (60.0)
HPV genotype, *n* (%)	
HPV-16	16 (18.0)
HPV-18	6 (6.9)
HPV-31	7 (8.0)
HPV-33	1 (1.1)
HPV-39	6 (6.9)
HPV-45	1 (1.1)
HPV-51	5 (5.7)
HPV-52	3 (3.4)
HPV-56	5 (5.7)
HPV-58	8 (9.2)
HPV-59	8 (9.2)
HPV-66	10 (11.5)
HPV-68	10 (11.5)
Cytology analysis, *n* (%)	
Negative	18 (29.5)
ASCUS	21 (34.4)
LSIL	22 (36.1)
Colposcopy examination, *n* (%)	
Negative	16 (32.0)
G0	4 (8.0)
G1	20 (40.0)
G2	10 (20.0)
Histology examination, *n* (%)	
Negative	16 (57.1)
CIN1	8 (28.6)
CIN2	3 (10.7)
CIN3	1 (3.6)
Methylation test positivity, *n* (%)	12 (19.7)

**Table 2 cancers-16-01986-t002:** Sensitivity, specificity, positive, and negative predictive values of DNA methylation versus cytology, HPV-DNA, colposcopy, and histology analysis.

DNA Methylation	Cytology (*n* = 61)		*p*-Value	HPV-DNA (*n* = 61)		*p*-Value	Colposcopy (*n* = 50)		*p*-Value	Histology (*n* = 28)		*p*-Value
	*Pos*	*Neg*		*Pos*	*Neg*		*Pos*	*Neg*		*Pos*	*Neg*	
Positive, *n*	10	2	*0.48*	12	0	*0.59*	10	2	*0.09*	2	3	*0.14*
Negative, *n*	33	16		43	6		20	18		2	20	
Sensitivity (%)	23.3		-	22.0		-	33.3		-	50.0		-
Specificity (%)	88.9		-	100.0		-	90.0		-	87.0		-
PPV	0.83		-	1.00		-	0.83		-	0.40		-
NPV	0.33		-	0.12		-	0.47		-	0.90		-

**Table 3 cancers-16-01986-t003:** Sensitivity, specificity, positive and negative predictive values of cytology versus HPV-DNA, colposcopy, and histology analysis.

Cytology	HPV-DNA (*n* = 61)		Colposcopy (*n* = 50)		Histology (*n* = 28)	
	Pos	Neg	Pos	Neg	Pos	Neg
**Positive**	38	5	26	9	4	18
**Negative**	17	1	4	11	0	6
**Sensitivity (%)**	69.1		86.7		100.0	
**Specificity (%)**	16.7		55.0		25.0	
**PPV**	0.88		0.74		0.18	
**NPV**	0.05		0.73		1.0	

**Table 4 cancers-16-01986-t004:** Sensitivity, specificity, positive and negative predictive values of HPV, and methylation testing combined versus colposcopy results.

	**Colposcopy**		
		*POS*	*NEG*
**Co-testing HPV-DNA+/Methylation+**	*POS*	8	4
	*NEG*	22	16
**Sensitivity (%)**	31%		
**Specificity (%)**	80%		
**PPV**	0.71		
**NPV**	0.42		

**Table 5 cancers-16-01986-t005:** Clinical performance of FAM19A4/mir124-2 methylation marker analysis, HPV16 genotyping, and the combination of both triage tests for positive colposcopy outcome (i.e., ≥1 grade lesions).

Triage Test	Sensitivity (%)	Specificity (%)	PPV (%)	NPV (%)
DNA Methylation	33	90	83	47
Cytology	87	55	74	73
HPV-16 positive	37	75	69	44
Methylation+/Cytology+	27	90	80	45
Methylation+/HPV+	27	80	67	42

## Data Availability

The datasets used and/or analyzed during the current study are available from the corresponding author upon reasonable request.

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
