# Peer review of "Diagnostic Accuracy of DNA-Methylation in Detection of Cervical Dysplasia: Findings from a Population-Based Screening Program"

_cancers, 2024, doi:10.3390/cancers16111986_

Round 1
Reviewer 1 Report
Comments and Suggestions for Authors
In this manuscript, the authors evaluated the performance of DNA methylation and currently used cervical cancer screening techniques. The authors found a good specificity of DNA methylation tests and indicated the FAM19A4/miR124-2 as reliable biomarkers in cervical cancer screening. However, the analysis lacks statistical tests, and the manuscript needs careful improvement and revision. Here are my specific comments:
1. The analysis included only a limited number of samples, and no statistical tests were performed to assess significance. For all data presented in Table 2 (line 205) and similar analysis, a chi-squared test or Fisher’s exact test should be implemented to assess whether there is a significant association between DNA methylation and other test results.
2. Due to the lack of statistical analysis, it’s difficult to assess the potential role of DNA methylation in cervical cancer screening.
3. The Table 1 on line 205 should be Table 2.
4. The number of cytology tests reported in Table 2 is inconsistent with Figure 3.
5. The combination of methylation test with HPV-DNA result led to an increase in the AUC from 0.53 to 0.54, which does not represent a significant improvement.
Comments on the Quality of English Language
Minor suggestions for improvement.
Author Response
Please, see the attached file.

Reviewer 2 Report
Comments and Suggestions for Authors
The authors have investigated the diagnostic and clinical performance of FAM19A4/miR124-2 methylation-based triage for severe cervical dysplasia by comparing the results of methylation test with previous cytological, colposcopy and histological examinations, in a population-based cervical screening program. The samples were randomly selected regardless of the HPV-, cytology- or and histology- result. This makes the approach interesting because it reflects population based screening. On the other hand the approach has an inherent weakness because only a 4 of 61 patients actually had CIN2/3. Thus, the sensitivity analyses of the various markers as shown in Table 4 are of limited meaning.
Minor points:
1. Figure 1 should be omitted since the same data is already listed in Table 1.
2. In Table 1 a fine horizontal line should be drawn between "Histological exam" and "Methylation testing".
3. In the results section (line 232) should read HPV-DNA not HPD-DNA.
4. The discussion could be shortened. The paragraph beginning with line 258 could be omitted.
Author Response
Please, see the attached file.
